# The Malay Literacy of Suicide Scale: A Rasch Model Validation and Its Correlation with Mental Health Literacy among Malaysian Parents, Caregivers and Teachers

**DOI:** 10.3390/healthcare10071304

**Published:** 2022-07-14

**Authors:** Picholas Kian Ann Phoa, Asrenee Ab Razak, Hue San Kuay, Anis Kausar Ghazali, Azriani Ab Rahman, Maruzairi Husain, Raishan Shafini Bakar, Firdaus Abdul Gani

**Affiliations:** 1Department of Psychiatry, School of Medical Sciences, Universiti Sains Malaysia Health Campus, Kubang Kerian 16150, Kelantan, Malaysia; picholas@student.usm.my (P.K.A.P.); huesankuay@usm.my (H.S.K.); drzairi@usm.my (M.H.); raishanshafini@usm.my (R.S.B.); 2Biostatistics and Research Methodology Unit, School of Medical Sciences, Universiti Sains Malaysia Health Campus, Kubang Kerian 16150, Kelantan, Malaysia; anisyo@usm.my; 3Department of Community Medicine, School of Medical Sciences, Universiti Sains Malaysia Health Campus, Kubang Kerian 16150, Kelantan, Malaysia; azriani@usm.my; 4Department of Psychiatry and Mental Health, Sultan Haji Ahmad Shah Hospital, Temerloh 25000, Pahang, Malaysia; drfirdaus.gani@moh.gov.my

**Keywords:** psychometric analysis, Rasch model analysis, suicide literacy, mental health literacy, adolescent mental health

## Abstract

The 27-item Literacy of Suicide Scale (LOSS) is a test designed to measure the respondent’s suicide knowledge. The purpose of this study is to examine the psychometric properties of the Malay-translated version of the LOSS (M-LOSS) and its association to sociodemographic factors and mental health literacy. The 27-item LOSS was forward–backward translated into Malay, and the content and face validities were assessed. The version was distributed to 750 respondents across West Malaysia. Rasch model analysis was then conducted to assess the scale’s psychometric properties. The validated M-LOSS and the Malay version of the Mental Health Knowledge Schedule (MAKS-M) were then distributed to 867 respondents to evaluate their level of suicide literacy, mental health literacy, and their correlation. Upon Rasch analysis, 26 items were retained. The scale was found to be unidimensional, with generally satisfying separation and reliability indexes. Sex, socio-economic status, and experience in mental health were found to significantly impact the mean score for mental health literacy. This study also found a significant mean difference for suicide literacy across school types. Furthermore, while this study observed a weak but significant negative correlation between age and suicide literacy, no correlation was found between mental health and suicide literacy.

## 1. Introduction

The COVID-19 pandemic forced the closure of all educational institutions, from pre-primary to tertiary institutions, in 200 countries. This closure halted face-to-face teaching and learning and affected 1.58 billion students worldwide. It is deemed the largest educational interruption in history [1]. While the prospect of learning from the comfort of our own home initially sounded appealing, the negative impacts of the biodisaster on mental health slowly surfaced, including those on children and adolescents. The social isolation, lack of certainty, fear of infection, and increased risk of household violence could lead to increased anxiety, depression, stress, and adjustment disorders, which could exacerbate existing mental health problems [2]. Compounding the issue, mental health issues such as depression and anxiety are associated with a higher risk of suicidal behaviors [3]. Since the outbreak, a recent study in the US found an increase in positive screening for suicide risk among adolescents [4]. A similar increasing trend in the number of suicide cases was also observed in Malaysia—from 609 cases in 2019 to 631 cases in 2020—while within the first quarter of 2021, a total of 336 suicide cases were reported, which is a matter of serious concern [5].

Mayne et al. also highlighted the suspension of mental health screening activities and in-person visits during the pandemic, possibly aimed at reducing clinic visits to lower the risk of COVID-19 infection, causing a decreasing trend on depression screening at primary care visits [4]. In this situation, adult stakeholders, i.e., parents, caregivers, and teachers, play an important role in identifying mental health issues and providing initial psychological first aid for adolescents in mental distress. Parents and caregivers were the second-most preferred source of mental health assistance among adolescents after their peers due to the presence of trust and interpersonal relationships [6,7]. Although teachers were not the preferred source of help, teachers were more likely to encounter teens suffering from mental illnesses, since they were required to meet and communicate with children and adolescents for most of their working hours. A qualitative study among parents and caregivers of adolescents suffering from mental illnesses stated that most of their children’s mental health problems were initially detected by school personnel [8].

Of late, mental health services are pressured to enhance suicide risk assessment to lower the suicide rate. As a result, suicide is mostly seen as a psychiatric illness [9]. In this light, while virtually all mental illnesses are associated with an increased risk of suicide, suicide does not always indicate an underlying mental illness. A review article stated that up to 66.7% of suicide cases were absent of Axis 1 disorders, 37.1% did not have underlying personality disorders and Axis 1 disorders, and 37% of suicide cases did not have underlying Axis 1 disorders with mild conditions [10]. In high-income nations, psychological autopsy revealed that up to 10% of those who died by suicide did not have an underlying mental illness. The rates were higher in several Asian countries [11]. Additionally, several studies found that adverse childhood experiences may lead to suicide attempts among the youth, mediated by maladaptive personality development (e.g., aggression, impulsivity, learning difficulties, substance abuse, and chronic pain) [12,13]. While mental disorders and suicide are the most explored areas in suicidology, we have yet to come across studies that prove the link between suicide literacy and mental health literacy. This information might be useful in shaping future mental health and suicide awareness campaigns, ensuring that both issues receive equal attention.

Improving mental health and suicide literacy is an important first step in understanding the issues, reducing stigma, and enhancing confidence in providing help [14]. In Malaysia, studies on suicide literacy were rather limited to medical- and healthcare-related professionals and students [15,16,17]. To the best of our knowledge, there have been no studies on suicide literacy among parents, caregivers, and teachers. Thus, there are no available baseline data for these population subsets. To assess the level of knowledge on suicide, the Literacy of Suicide Scale (LOSS) was developed [18]. The scale is dichotomous and consists of 27 items encompassing the signs and symptoms, risk factors, causes, nature, treatment, and prevention of suicide. The Mental Health Knowledge Schedule (MAKS), on the other hand, is a two-part instrument consisting of 12 items that were designed to evaluate several dimensions of mental health literacy. This study aims to translate LOSS into Malay and validate it among Malaysian parents, caregivers, and teachers of secondary school adolescents. This study also aimed to compare the mean scores of mental health literacy and suicide literacy levels among the subgroups and investigate the correlations between sociodemographic factors, mental health literacy, and suicide literacy among the population. The study hoped to assess suicide literacy among the Malay-speaking population to quantify their level of understanding on suicide and evaluate the effectiveness of suicide awareness interventions. Investigation into the relationship between sociodemographic factors and the level of mental health and suicide literacies also allowed the conceptualization of factors that affect the maintenance of mental health and informs the policy and strategy development for mental health and suicide awareness interventions to encourage positive help-seeking behavior and confidence in providing appropriate mental assistance.

## 2. Materials and Methods

### 2.1. Study Design and Subjects

This study employed a cross-sectional study design. The targeted population comprised parents, caregivers, and teachers of adolescents in secondary school. The source population is the parent–teacher association of secondary schools in West Malaysia. The schools were selected via multistage stratified random sampling. First, government secondary schools in West Malaysia were stratified based on the type of schools (i.e., national secondary schools “SMK”, fully residential schools “SBP”, and religious secondary schools “SMA”) and the locality of school (i.e., urban and rural). States in West Malaysia were then divided into four geographical zones (i.e., northern, southern, eastern, and central zones). For each stratum, simple random sampling was used to select one state from each zone and one district from each state. Finally, one school was randomly chosen from each district. A total of 24 schools were involved in this study. A link to the questionnaire and a poster with the QR code were circulated among the target sample via a key informant from each school’s parent–teacher association for recruitment. The target sample comprised parents or caregivers of students in these schools and teachers who were still actively teaching in the selected schools. The inclusion criteria included Malaysian, aged 18 and above, and literate in Malay. Individuals who could not read in Malay were excluded from this study. The data were collected between March and September of 2021.To conduct a Rasch model analysis, a minimum sample size of 250 respondents, or about 25 respondents per response category, is necessary to achieve stable item calibration [19]. For this cross-sectional study, single proportion formula was adopted with 80% statistical power and at a significance level of 5%. We expected a dropout rate of 20%, and a design effect of 2 was applied, yielding an effective sample size of minimum 730 respondents.

### 2.2. Instrument

LOSS is a 27-item scale constructed to assess the knowledge on suicide among the respondents based on four categories: (a) cause and nature (CN), (b) risk factors (RF), (c) signs and symptoms (SS), and (d) treatment and prevention (TP) of suicide. Since LOSS is a dichotomous scale, respondents were required to choose “true” or “false” according to their understanding of suicide. Respondents were given 1 score for each correct answer, whereas 0 scores were given for incorrect answers. The total score was obtained by summing the scores of items answered correctly. The score ranged between 0 and 27; the higher the score, the higher the suicide literacy. During the development, the LOSS was validated via the item response theory approach. Each item was constructed based on the latent trait rather than factor analysis and internal consistency. In this light, several past studies that utilized LOSS have reported an acceptable Cronbach’s alpha value of 0.71 [18,20].

MAKS is an instrument designed to assess general mental health knowledge. The schedule comprises 12 items in two parts; Part A evaluates knowledge of mental health stigma regarding help-seeking, recognition, support, employment, treatment, and recovery. Part B evaluates knowledge on mental health diagnoses. Items in MAKS are scored based on an ordinal scale between 1 to 5, and the response “Don’t know” is represented by 3 on the scale, similar to “Neutral”. The sum of scores ranged between 12 to 60, and a higher score indicates better mental health literacy. The original version of MAKS was found to have moderate to substantial internal reliability for Part A (alpha = 0.65) and test–retest reliability of 0.71 [21]; however, since the instrument purposely included multidimensional items, the low internal reliability was expected and permissible for testing various types of stigma knowledge. The current study adopted the Malay version of MAKS (MAKS-M), which was translated and adapted for Malaysian secondary school teachers and yielded an internal consistency of 0.62 [22]. The internal consistency for Part A and Part B of the scale were 0.54 and 0.71, respectively. Overall, the Cronbach’s alpha of MAKS-M within this study’s population was 0.68, which is consistent with the studies mentioned above.

### 2.3. Procedure

#### 2.3.1. Translation and Validation of the LOSS

The Malay version of the LOSS (M-LOSS) was translated using the forward–backward translation method. The English version of LOSS was translated into Malay by two researchers. Subsequently, after discussion and reconciliation, the two independent forward translations were merged into one interim translation. The backward translation from Malay to English was performed by an independent translator who had not read the original English version of the scale. The backward translation of the scale was then compared to the original version. Alterations were made accordingly until a consensus Malay version was produced. This harmonization was then distributed among ten Malaysian adult subjects whose first language is Malay to conduct a cognitive debriefing. Several pieces of feedback were obtained, and amendments were made accordingly. The researchers then proofread the final version of the Malay-translated questionnaire to correct minor errors before the validation process.

The M-LOSS was then distributed to six experts—two psychiatrists, one community medicine specialist, one family medicine specialist, one psychologist, and one biostatistician—for content validation. The panel of experts rates each item based on the level of relevance based on a Likert scale ranging from 1 (i.e., not relevant) to 4 (i.e., highly relevant). The ratings were entered into Microsoft^®^ Excel for Mac V.16.55 (Microsoft, Redmond, WA, USA) and processed with the content validity index (CVI) calculation. Items with the content validity index (I-CVI) value of 0.83 and above were retained [23,24,25].

The face validity of the M-LOSS was assessed by ten individuals—six parents and four teachers. All individuals are Malaysian citizens and were chosen via purposive sampling by the researchers. This step was performed via Google Form^®^ (Google, Mountain View, CA, USA) survey. The individuals were required to rate the items based on clarity and comprehensibility. The ratings were based on a 4-point Likert scale from 1 (i.e., not clear and understandable) to 4 (i.e., very clear and understandable). The responses were gathered and calculated using Microsoft^®^ Excel for Mac V.16.55 (Microsoft, Albuquerque, NM, USA) to obtain the face validity index (FVI). Items with an item level face validity index (I-FVI) of 0.83 or above were retained [26,27].

The questionnaire was then pilot tested among 139 parents and teachers using online Google Form^®^ (Google, Mountain View, CA, USA). Through snowball sampling, the participants were selected among parents and teachers of secondary school students within the Kelantan state. The pilot test aimed to obtain feedback on administrative procedures, such as the instrument’s timing and structure before data collection.

The 27-item M-LOSS was circulated among parent–teacher associations of selected schools via one key informant. The schools were sampled via multistage stratified random sampling as described previously. Brief information regarding the study was provided at the beginning of the form. Consequently, informed consent was obtained by clicking the option “agree” to enable them to proceed to subsequent sections. Once the validation was completed, the validated M-LOSS, the MAKS-M, and sociodemographics were circulated among the schools via a similar method as mentioned previously. All completed responses were automatically saved into the Google Form^®^ database (See Figure 1: Study flowchart).

#### 2.3.2. Statistical Analysis

The psychometric properties were evaluated via WINSTEPS^®^ 3.68.2 (Linacre, USA) to analyze the construct validity through Rasch model analysis. Several important aspects of Rasch’s analysis are outlined below.

Item polarity or point measure correlation (PTMEA Corr.) was performed to observe the correlations between the items within the scale for initial detection of the instrument’s construct validity. A positive correlation indicates the items are pointing in the same direction, and good PTMEA Corr. should be 0.20 and above [28].

This study used inlier-pattern-sensitive fit statistics (INFIT) and outlier-sensitive fit statistics (OUTFIT) for Chi-squared-based tests of the model’s fit for the item fit analysis. INFIT detects unexpected responses to items based on their ability level, whereas OUTFIT considers any discrepancies between observed and expected responses, disregarding the differences between the item difficulty and personal ability measures [29]. Item fit analysis is reported using mean squared (MNSQ) and z-standardized (ZSTD) scores, for which MNSQ is the Chi-squared calculation of fit statistics, and the ZSTD measures the probability of the MNSQ calculation occurring by chance. A productive item that fits into the model will have INFIT and OUTFIT MNSQ values within the range of 0.5 to 1.5. Meanwhile, if the ZSTD value exceeds 2.0, the item is considered erratic. On the other hand, if the MNSQ value is within the optimal range, the ZSTD value can be omitted, since the value is dependent on the MNSQ [30].

Principal component analysis of residual (PCAR) was used to test the dimensionality of the instrument, since unidimensionality is one of the assumptions in Rasch model analysis. Unidimensionality is met if the amount of variance explained by measures is more than 20%, the unexplained variance of the eigenvalue for the first contrast is less than 3, and the unexplained variance accounted for by the first contrast is less than 5% [31].

Item and person separation indexes were examined to separate the items, based on difficulty, and the respondents into several groups. The scales were expected to distinguish at least two groups of people or items. Therefore, the acceptable minimum value for the separation index is 1.50 [32]. This study also assessed the item and person reliability indexes. Item reliability index indicates if identical items’ placements can be replicated if given to another sample with a similar skill level. In the meantime, the person reliability index indicates that a person’s ranking could be replicated if he/she were given another set of items that evaluate similar constructs [33,34]. The reliability index ranged between 0 to 1. In this study, the acceptable value for the reliability index was set at 0.70 or above [32].

The Wright map visualizes the distribution of both the person measures and the location of the items. On the left of the map, person abilities are arranged from the highest to the lowest, whereas on the right, item difficulties are arranged from the hardest to the easiest from top to bottom.

Following the Rasch analysis, the descriptive data of the respondents’ sociodemographics were obtained using IBM^®^ SPSS^®^ Statistics 27.0 (IBM, Armonk, NY, USA). Normality was assumed in this study based on the central limit theorem, whereby as the sample size grows higher (at least more than 30 samples), the distribution of sample means approaches a normal distribution [34]. This assumption was further verified by the value of skewness and kurtosis for each variable. Sample characteristics were tabulated, and the mean scores of MAKS-M, M-LOSS, and each subgroup were compared using the independent t-test and one-way ANOVA after data cleanup. The Pearson correlation analysis examined the correlation between age, suicide literacy, and mental health literacy. Results with *p* < 0.05 are considered statistically significant.

### 2.4. Ethical Consideration

The university’s ethical committee has approved this study’s proposal and instrument. Furthermore, permission was obtained from the author to adopt the questionnaire in this study. Information sheets outlining the study purpose, procedure, eligibility criteria, possible risks and benefits, confidentiality, and researchers’ contact information were made available to the participants. Potential participants were informed that their participation was voluntary and that the data collected would be anonymous. After the full disclosure of the study, online informed consent was sought from individuals interested in participating in the study. Respondents who expressed their interest were redirected to the following sections in the Google Form.

## 3. Results

### 3.1. Psychometric Properties of M-LOSS

#### 3.1.1. Translation

The 27-item M-LOSS was translated using the forward–backward translation method as described above. After the questionnaire was translated, ten Malaysians who are Malay native speakers were involved in the cognitive debriefing session. One subject commented that the word “psikotik” (psychotic) in item 11 is a medical term and should be replaced with a simpler word. However, due to the lack of appropriate synonyms, examples of psychotic symptoms such as hallucination and delusion (i.e., berhalusinasi, delusi) were added instead. Other than that, only several minor changes involving grammatical errors were made.

#### 3.1.2. Content Validity

A total of 17 out of 27 items obtained universal agreement (UA), with an I-CVI value of 1.00. Items 6, 7, 9, 14, 18, 19, 20, 21, 22, and 23 obtained an I-CVI value of 0.83. Five out of six experts considered these items “relevant” or “very relevant”. In general, all items were retained, as they obtained I-CVI values of 0.83 and above, with an overall average CVI of 0.94. The results were compiled and further discussed amongst the researchers. No changes were made in this phase (see Appendix A Appendix A).

#### 3.1.3. Face Validity

Ten subjects were involved in the face validation study. A total of 20 out of 27 items achieved UA with an I-FVI value of 1.00. Items 3, 5, 6, 7, 8, 14, and 24 were rated “clear and comprehensive” or “very clear and comprehensive” by nine out of ten raters, resulting in an I-FVI score of 0.90. Hence, all items were retained, and no amendments were necessary (see Appendix A Appendix A).

#### 3.1.4. Pilot Test (*n* = 139)

The pilot test of M-LOSS involved 139 respondents (70 parents, 69 teachers). The respondents’ mean age was 40.99 ± 13.26 years old. The majority of respondents were female (70.5%), Malay (61.9%), practiced Islam (68.3%), received tertiary education (78.4%), and worked in the government sector (65.5%).

The respondents’ overall acceptance was positive. They required approximately 5–10 min to complete the questionnaire. This indicates that the online survey form was well-structured, and the respondents did not experience any technical difficulties. The items were appropriate to the topic, unambiguous, and easily understood. As a result, a 27-item M-LOSS on knowledge of suicide with dichotomous response option was produced.

#### 3.1.5. Rasch Model Analysis (*n* = 750)

Overall, 750 responses were received between March to June 2021. The respondents’ mean age and standard deviation were 41.61 (6.97). More than half of the respondents were parents/caretakers (65.5%), whereas the others were teachers (34.5%). Most of the respondents were female (73.6%), Malay (88.0%), and practiced Islam (90.7%). A large portion of the respondents had completed tertiary education (76.7%) and were working in the government sector (68.1%). A majority of the respondents come from an urban school setting (58.3%) compared to a rural school setting (41.7%). There was an almost equal percentage of participants from national secondary schools (39.6%) and full boarding schools (37.2%), while there were fewer participants from religious secondary schools (23.2%).

In terms of item polarity, the first Rasch model analysis of the 27-item M-LOSS found that only one item, Item 18, had a low correlation (PTMEA Corr. = 0.16) (see Appendix A Appendix A). The item was removed, and the analysis was repeated with the 26-item M-LOSS. The remaining 26 items were found to have positive correlations ranging between 0.20 and 0.46. As for the INFIT and OUTFIT item fit analyses, the MNSQ values for all 26 items were within the optimal range, varying between 0.90 to 1.11 for the INFIT MNSQ and between 0.83 to 1.17 for the OUTFIT MNSQ. The ZSTD scores were ignored, since all the MNSQ values were acceptable (see Appendix A Appendix A). Thus, all the remaining 26 items fit the model and could be used for this study (see Appendix B Table A1).

As for the PCAR, the amount of raw variance explained by the measures is 29.0%, and the amounts of raw variance explained by persons and items were 10.7% and 18.4%, respectively. Although the unexplained variance in the first contrast was 5.3%, which marginally exceeded the cut-off value by 0.3%, the eigenvalue of the first contrast was 2.0, which met the requirement of <3.0. Therefore, the data fit the Rasch model, as proven by the unidimensionality of the instrument (see Appendix A Appendix A).

The person separation index for this current sample is 1.52, which suggests that the 26-item M-LOSS can separate the respondents into at least two strata based on their ability. The item separation index was 12.47, indicating an excellent level of separation. The person reliability index achieved the minimum acceptable value of 0.70, whereas the item reliability was near-perfect at 0.99. Overall, the separation and reliability indexes for the 26-item M-LOSS instrument were satisfactory for both person and item measures (see Appendix A Appendix A).

The Wright map charts the person and item measures based on their ability and difficulty level. In Figure 2, a bell-shaped distribution can be seen for the person measures (left), with the respondents at the top having higher literacy on suicide and vice versa. The item measures (right) show that Items 27, 5, 3, and 12 are the easiest and do not coincide with any respondents’ ability levels. Conversely, Items 24 and 26 are the two most difficult items within the scale. However, these items are still within the ability range of the respondents. There are several items at a similar position, especially within the +1 to +2 logits (e.g., Items 17, 25, 7, and 8), signifying a possible redundancy from the measurement perspective. Finally, a gap of more than a logit was found above Items 24 and 26.

### 3.2. Suicide Literacy and Mental Health Literacy (*n* = 867)

A total of 882 responses were collected between July to September 2021. After the data cleanup, 867 responses were retained for further analysis. The respondents’ mean age (SD) was 43.80 (8.35) years. The majority of the respondents were parents and caretakers (64.6%), female (71.5%), Malay (87.1%), and practiced the Islamic faith (89.9%). Three quarters of respondents had completed tertiary education (75.2%), were working in the government sector (66.7%), and had a household income within the middle 40% (M40) income bracket (45.4%). The respondents were mostly from SMK (39.3%) in urban areas (58.1%). Only several respondents self-reported a personal history of mental health (2.4%), had had known contact with mental health issues (18.5%), had experience in assisting those with mental health issues (25.0%), or had attended formal mental health first aid training (10.1%) (see Appendix A Appendix A).

The mean scores for MAKS-M and M-LOSS for the 867 parents, caregivers, and teachers are tabulated in Table 1 and Table 2, respectively. The overall mean score (SD) for MAKS-M is 43.85 (SD = 4.07); the mean score for Part A is 21.36 (SD = 2.55), and that for Part B is 22.46 (SD = 2.66). For M-LOSS, the respondents’ overall mean score is 14.05 (SD = 2.61), equivalent to 54.0%. Respondents obtained the lowest score in CN (42.3%), followed by SS (53.4%) and RF (60.4%), and the highest score was in TP (72.5%).

The independent t-test and one-way ANOVA results for MAKS-M are presented in Table 3. The analysis revealed significant differences in knowledge on mental health stigma mean scores based upon sex (*p* = 0.018), knowing someone with mental disorders (*p* < 0.001), having experience in assisting psychiatric patients (*p* = 0.002), and having attended formal training in psychological first aid (*p* = 0.038).

Additionally, religion (*p* = 0.047), education level (*p* = 0.007), household income bracket (*p* < 0.001), knowing someone with psychiatric illness (*p* = 0.024), and having attended formal psychological first aid training (*p* = 0.021) were also found to have significantly different mean scores for knowledge on mental health diagnoses.

Finally, factors that have a significant mean difference in the overall mental health literacy include the level of education (*p* = 0.003), household income (*p* = 0.012), close contact with psychiatric patients (*p* < 0.001), experience in assisting psychiatric patients (*p* = 0.019), and formal training in psychological first aid (*p* = 0.008).

Further analysis using the Tukey–Kramer post hoc test for the level of education suggested that only the subgroup of respondents receiving primary education and tertiary education were found to be significantly different (*p* = 0.017). Respondents with tertiary-level education were found to score better in mental health diagnoses, with a mean difference of 3.639. A post hoc test for household income demonstrated a significant difference in the mean score for mental health diagnoses between B40 and M40 households (*p* = 0.002), with mean difference of 0.672, and B40 and T20 households (*p* = 0.002), with a mean difference of 0.887. The mean score differences for overall mental health literacy between respondents from B40 and M40 were significant (*p* = 0.040), with a mean difference of 0.732, and B40 and T20 income brackets were also statistically significant at *p* = 0.026, with a mean difference of 1.122. Post hoc analysis on religion revealed no significant pairwise mean difference between each subgroup despite the significant one-way ANOVA.

For the suicide literacy, we reported a significant mean difference between the school type and the knowledge on SS (*p* = 0.003) and the overall knowledge on suicide (*p* = 0.035); the post hoc test revealed a difference in mean score between respondents from SBP and SMA for mean scores of SS (*p* = 0.002) with a mean difference of 0.296 and overall suicide literacy (*p* = 0.037) with mean difference of 0.574. There is no statistically significant difference for SMK compared to SBP or SMA. The results are tabulated in Table 4.

Finally, Pearson’s correlation analysis findings are tabulated in Table 5. The correlation between age and M-LOSS is significant and negatively correlated (r = −0.076, *p* = 0.025). However, the strength of correlation was deemed weak. The finding also indicates that other variables are not significantly correlated.

## 4. Discussion

### 4.1. Rasch Model Analysis of 26-Item M-LOSS

The current study aimed to investigate the psychometric properties of the M-LOSS using Rasch model analysis to develop a reliable and validated instrument to measure the level of suicide knowledge among the Malaysian population. Item polarity and item fit analysis suggested that one item (Item 18, “not all people who attempt suicide plan their attempt in advance”) has a relatively low correlation. Although the correlation value of Item 18 was positive, indicating the item was aligned with the overall measure, the relatively low correlation value denotes possible inconsistency of indicator polarity in the scale. This indicates the item is not working well with the others. Consequently, Item 18 was removed [28]. Otherwise, the remaining 26 items showed positive PTMEA Corr. values within the acceptable range with stable INFIT and OUTFIT MNSQ values, signifying that all items fit the Rasch model and are appropriate to the instrument measure.

As for the separation and reliability indexes, the study achieved satisfactory values for both the person and item measures. In this study, based on the 26-item M-LOSS, a minimum of two-person strata can be generated using this instrument. Furthermore, adequate person reliability indicates that the items are consistent in establishing hierarchy based on the person’s ability. The excellent item separation and reliability, on the other hand, imply that the items developed are varied in difficulty level while maintaining consistency and replicability of item placement across different samples [28].

For the spread of items in the Wright map, a gap of more than one logit difference was detected above Item 24. This gap may be caused by the borderline person separation index, since no item of appropriate difficulty coincided with the respondents’ ability at such level. Thus, the scale was unable to stratify individuals with such level of ability. For improvement, more difficult items can be included to better assess and differentiate respondents with high abilities [28]. On the other hand, the cluster of items between the +1 to +2 logits is deemed psychometrically redundant (e.g., Items 17, 25, 7, and 8), as these items measure a similar ability level. Although these items can be dropped, we argue their importance, as they measure different categories of suicide knowledge. For instance, Items 5 and 3 evaluate the causes and nature of suicide, Item 12 assesses the risk factor, while Item 27 assesses the treatment and prevention of suicide [18]. Bond et al. also added that psychometric and theoretical redundancy are unalike perspectives. Therefore, the initial item development and rationalization should be taken into consideration [28].

### 4.2. Mental Health Literacy and Suicide Literacy

The current study population scored an average mental health literacy level of 43.82. This is comparable to a separate study among Malaysian teachers by Tay et al., with a mean score of 42.32 [35]. The study sample scored relatively higher than the Lebanese community but lower than studies in Jordan and the United Kingdom [36,37,38]. It is worth noting that the latter two studies recruited caregivers of patients with psychosis and healthcare workers for their studies, in which they not only worked closely with patients with underlying psychiatric conditions but also received formal training and psychoeducation on various mental health issues. This could explain their population’s higher mental health literacy level compared to ours. Although the majority of Malaysian and Lebanese share an identical Islamic theological view on mental health (mental illnesses are associated with sins, divine punishments, and demonic possession), it is argued that Malaysian Muslims are more supportive of psychiatric patients and consider the illness as an opportunity to strengthen their connection with God, as compared to Arabic Muslims, for whom mental illnesses are perceived as shameful and disgraceful to the family [36,39,40].

For the suicide literacy, our sample scored only 54.0% correct responses, which is slightly lower than the mean score of community samples worldwide (58.2%) [41]. Still, this result is deemed comparable with the scores obtained from other populations—South Indian (50.9%), Chinese (53.0%), and Jordanian (55.0%) [20,42,43]. Several other studies reported lower rates of suicide literacy among Turkish (36.9%) and Bangladeshis (43.3%) [44,45] and higher scores among the Australian (>60.0%) and German (58.3%) populations [18,46,47]. This could be due to the lack of emphasis on mental health and suicide prevention programs in emerging and developing countries (EDCs) as opposed to developed countries (i.e., Germany and Australia). Additionally, the EDCs focus on other emerging issues on human development (e.g., poverty, illiteracy, overpopulation), communicable diseases, and maternal and child health, consequently deprioritizing the mental health sector [48]. However, several studies on suicide literacy have reported a similar trend of items under the TP theme being items with the most correct answers, while other subthemes scored lower [41]. This result is also reflected in this study. This may be due to the low item difficulty or a knowledge gap between different domains of suicide. Ludwig et al. further reasoned the items within the TP subtheme were too general and easy to identify [47]. This is thought to suffice, as the general population should be aware of the mental health services available and are not expected to have clinical knowledge such as psychotherapy and pharmacology. Nevertheless, the poor responses in the remaining subthemes are a call to highlight the nature, etiologies, risk factors, and warning signs of suicide in future suicide awareness and prevention programs to enable early recognition and intervention of suicidal individuals.

Individuals familiar with psychiatric illnesses and their interventions (i.e., had close contacts with psychiatric illnesses, had experience in assisting those with psychiatric illnesses, and attended formal psychological first aid training) were found to have significantly higher mean scores of mental health literacy than those who are not. This outcome is aligned with the findings by Doumit et al., wherein personal experiences enabled individuals to familiarize themselves with the symptoms and signs of mental health illnesses and recognize the appropriate diagnoses [36]. The caregiving experience also improved empathy and compassion. In turn, this helps improve knowledge on mental health stigma and reduce prejudice against those suffering from mental disorders [38].

Moreover, we also identified respondents having tertiary education as having scored better in mental health diagnoses and as having better overall mental health literacy than those who completed only primary level education. Similarly, those within the B40 income bracket scored significantly lower than their counterparts. Our findings support a previous study that identified higher education level and socioeconomic status as significant predictors for better mental health literacy [38]. These findings suggest a gap in accessibility for mental health awareness programs between different education levels and socioeconomic statuses. Individuals with higher education levels have more opportunities and exposure to mental health education and professional services, leading to more positive views on mental health issues and familiarity with diagnoses. Hence, more effort should be placed to improve mental health literacy among those with lower education levels from low-income households.

Our study also found that women have higher mental health stigma knowledge than men. Similarly, several other studies found that women show less stigma than men [36,49]. Doumit et al. argued that women are more empathetic and optimistic towards psychiatric patients, which could lead to such an observation [36]. On the other hand, most studies did not find any significant differences in mental health stigma between sexes. While both sexes exhibit mental health stigma, a qualitative study revealed that men have negative beliefs about diagnosis and treatment, whereas women’s stigmas heavily focus on society’s perceptions [50]. The varying outcomes between sex and mental health stigma knowledge warrant more research to shed light on these mixed findings.

For suicide literacy, the mean scores for the SS subtheme and overall suicide literacy among parents, caregivers, and teachers from SBP are significantly higher than those from SMA. A possible explanation is that boarding school students face additional risks of mental health issues such as homesickness, inadequate social support, and higher academic pressure. This could eventually lead to a higher prevalence of depression, anxiety, and stress among boarding school students [51,52]. Thus, parents and teachers of SBP are more aware of the warning signs of suicide for early detection of suicidal teenagers.

Religion could also be a barrier in discussing suicidality among SMA parents, caregivers, and teachers. Compared to other religions, Islam is firmer regarding the sins of suicide and self-harm, leading to increased social stigma and them being consequently regarded as taboos that should not be discussed openly among the community [53]. However, we have yet to identify any studies investigating how school types could influence the parents’, caregivers’, and teachers’ reception of mental health literacy programs. Thus, this is an interesting proposal to investigate in the future to further clarify our findings.

The study also observed a significant yet negligible negative correlation between age and the level of suicide literacy. The association between age and suicide literacy remained incongruent in other studies. A study by Öztürk reported no significant association between the two variables, while several other studies reported a significant association between age and suicide literacy [44,46,47]. They reasoned that due to the increased exposure to suicide awareness programs and openness in discussing suicide, the younger generation embraces the topic of suicide better than the older generation. However, younger people were thought to have limited understanding and minimal encounters with mental illnesses and suicide, leading to lower knowledge [43]. In this light, further research is required to reconcile the apparent disparity between this information.

Interestingly, no significant correlation was found between mental health literacy and suicide literacy. The current suicide prevention strategy is said to be too focused on treating mental disorders due to the high association of mental disorders and suicide [54]. Suicide prevention strategies should not only concentrate on mental health treatments but also include a diverse range of information to further enhance specific knowledge on suicide. For example, Mishara and Chagnon outlined that suicide is not only the direct consequence of mental illnesses (e.g., cognitive distortion, delusions, and psychotic command hallucinations), but other possible causes include the consequence of living with mental disorders (e.g., social stigma, hopelessness, social and relationship issues, dependency, unemployment, etc.), treatment inadequacy and its possible adverse effects, and additional crisis [54]. This calls for a need for adolescents diagnosed with mental disorders, their parents, caregivers, and teachers to be educated on suicide prevention, since they are at higher risk of suicide. Batterham et al. reported the clinical samples were found to have lower suicide literacy compared to the community sample, indicating that specific psychoeducation programs on suicide prevention for at-risk adolescents are required [55]. The lack of correlation between mental health and suicide literacy necessitates an emphasis on suicidality as separate psychoeducation, not just as an extension of mental health education.

### 4.3. Strengths and Limitations

This study is the first to translate and validate the LOSS questionnaire to evaluate suicide literacy levels among the Malaysian population and investigate its correlation with mental health literacy. This study’s multistage stratified random sampling enabled us to receive responses from a geographically dispersed population while maintaining adequate coverage across West Malaysia. However, our study faced several limitations. Firstly, we were unable to identify the exact response rate for this study due to the limited functionality in tracking the accessibility of the survey, which may put our study at risk of non-response bias. Furthermore, there is a scarceness in Malay-translated instruments to measure general mental health literacy among the Malaysian population. Although MAKS-M was reported to have less satisfactory reliability, it is a promising tool that enables the evaluation of stigma-related mental health knowledge and diagnoses. Therefore, improvement should be made to enhance the psychometric properties of this instrument. Furthermore, the interpretation of the LOSS score is reliant on comparison with other studies, due to the limited studies on suicide literacy in Malaysia, we could not compare these findings with other local populations. We anticipate that the current findings may serve as a useful tool for assessing suicide knowledge among Malaysians and as a foundation for evidence-based development of mental health and suicide awareness programs.

## 5. Conclusions

The current study has translated and validated the 26-item M-LOSS among Malaysian parents, caregivers, and teachers using Rasch model analysis to evaluate their suicide knowledge. The instrument was unidimensional, with adequate separation and reliability indexes, and all items had an acceptable correlation of 0.20 or above. All 26 items fit the model with INFIT and OUTFIT MNSQ within the optimal range, resulting in the M-LOSS having an overall favorable psychometric property. Furthermore, there may be value in directing mental health and suicide literacy programs towards parents, caregivers, and teachers who are male, who have lower education levels and socioeconomic status, who are unfamiliar with mental disorders, who did not receive formal training on psychological first aid, and who are in religious secondary schools. Lastly, no significant correlation was found between suicide literacy and mental health literacy.

## Figures and Tables

**Figure 1 healthcare-10-01304-f001:**
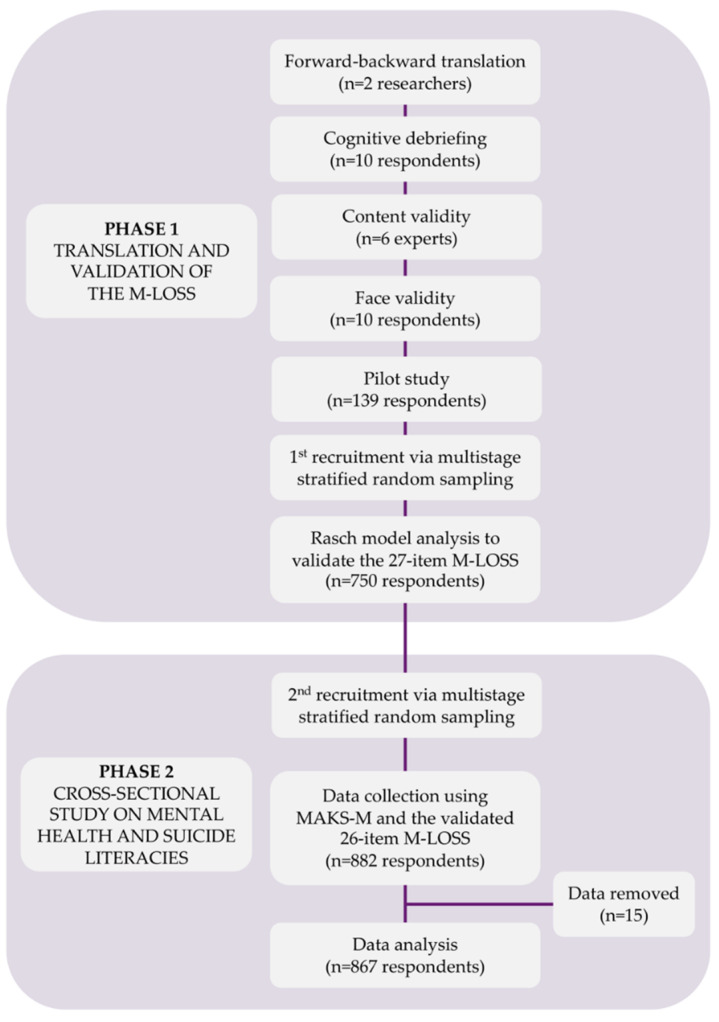
Study flowchart.

**Figure 2 healthcare-10-01304-f002:**
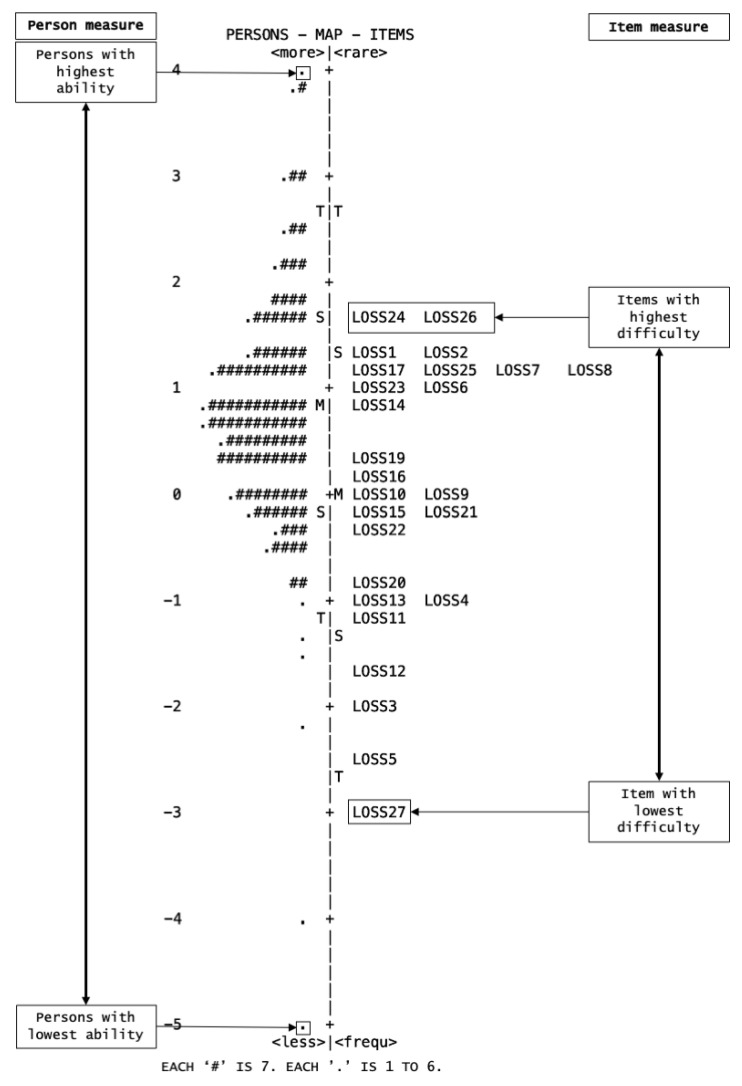
The Wright map for 26-item M-LOSS.

**Table 1 healthcare-10-01304-t001:** Responses frequency and mean scores for 12-item MAKS-M (*n* = 867).

Item	Part	Strongly Disagree n (%)	Disagree n (%)	Don’t Know/Neutral n (%)	Agree n (%)	Strongly Agree n (%)	Mean Score Mean (SD)
MAKS-M 1	A	16 (1.8)	149 (17.2)	294 (33.9)	259 (29.9)	149 (17.2)	21.36 (2.55)
MAKS-M 2		6 (0.7)	23 (2.7)	127 (14.6)	280 (32.3)	431 (49.7)	
MAKS-M 3		12 (1.4)	186 (21.5)	207 (23.9)	304 (35.1)	158 (18.2)	
MAKS-M 4		1 (0.1)	18 (2.1)	95 (11.0)	382 (44.1)	371 (42.8)	
MAKS-M 5		13 (1.5)	111 (12.8)	269 (31.0)	338 (39.0)	136 (15.7)	
MAKS-M 6		7 (0.8)	157 (18.1)	206 (23.8)	268 (30.9)	229 (26.4)	
MAKS-M 7	B	1 (0.1)	9 (1.0)	54 (6.2)	312 (36.0)	491 (56.6)	22.46 (2.66)
MAKS-M 8		7 (0.8)	45 (5.2)	97 (11.2)	299 (34.5)	419 (48.3)	
MAKS-M 9		1 (0.1)	15 (1.7)	141 (16.3)	288 (33.2)	422 (48.7)	
MAKS-M 10		2 (0.2)	16 (1.8)	133 (15.3)	295 (34.0)	421 (48.6)	
MAKS-M 11		34 (3.9)	125 (14.4)	143 (16.5)	313 (36.1)	252 (29.1)	
MAKS-M 12		15 (1.7)	75 (8.7)	138 (15.9)	366 (42.2)	273 (31.5)	
Overall mean score of mental health literacy							43.82 (4.07)

Note: SD = Standard deviation. Items are reverse scored.

**Table 2 healthcare-10-01304-t002:** Correct responses and mean scores for 26 item M-LOSS (*n* = 867).

Items	Themes	Correct Answer	Participants with Correct Answers n (%)	Mean Score Mean (SD)
M-LOSS 1	Causes and nature (CN)	False	526 (60.7)	4.23 (1.82)
M-LOSS 2		False	528 (60.9)	
M-LOSS 3		False	72 (8.3)	
M-LOSS 4		False	163 (18.8)	
M-LOSS 5		False	51 (5.9)	
M-LOSS 6		False	477 (55.0)	
M-LOSS 7		False	511 (58.9)	
M-LOSS 8		False	492 (56.7)	
M-LOSS 9		True	558 (64.4)	
M-LOSS 10		False	292 (33.7)	
M-LOSS 11	Risk factors (RF)	False	128 (14.8)	4.24 (1.24)
M-LOSS 12		True	782 (90.2)	
M-LOSS 13		True	700 (80.7)	
M-LOSS 14		True	430 (49.6)	
M-LOSS 15		True	603 (69.6)	
M-LOSS 16		True	528 (60.9)	
M-LOSS 17		False	505 (58.2)	
M-LOSS 18	Signs and symptoms (SS)	False	351 (40.5)	2.67 (0.97)
M-LOSS 19		True	683 (78.8)	
M-LOSS 20		False	284 (32.8)	
M-LOSS 21		True	608 (70.1)	
M-LOSS 22		True	386 (44.5)	
M-LOSS 23	Treatment and prevention (TP)	False	584 (67.4)	2.91 (1.07)
M-LOSS 24		False	520 (60.0)	
M-LOSS 25		False	584 (67.4)	
M-LOSS 26		True	839 (96.8)	
Overall mean score of suicide literacy				14.05 (2.61)

Note: SD = Standard deviation.

**Table 3 healthcare-10-01304-t003:** Comparison of mean between sociodemographic and MAKS-M (*n* = 867).

Variables	Part A Mean (SD)	Part B Mean (SD)	MAKS-M Mean (SD)
**Role** ^a^			
Parents and caregivers	21.31 (2.59)	22.33 (2.62)	43.65 (4.03)
Teachers	21.45 (2.48)	22.70 (2.71)	44.15 (4.13)
**Sex** ^a^			
Male	21.04 (0.16) *	22.30 (2.47)	43.34 (3.85) *
Female	21.49 (2.55)	22.53 (2.73)	44.02 (4.14)
**Ethnicity** ^b^			
Malay	21.39 (2.54)	22.53 (2.64)	43.91 (4.06)
Chinese	21.16 (2.51)	22.12 (2.65)	43.29 (3.65)
Indian	21.00 (2.16)	23.75 (2.06)	44.75 (4.11)
Other Bumiputera	21.50 (3.00)	21.63 (2.75)	43.13 (4.69)
Others	20.09 (1.38)	22.73 (2.83)	42.82 (3.55)
**Religion** ^b^			
Islam	21.40 (2.54)	22.53 (2.64) *	43.94 (4.06)
Christian	20.94 (2.63)	21.49 (2.69)	42.43 (4.18)
Buddhist	21.09 (2.75)	22.03 (2.68)	43.12 (3.91)
Hindu	21.67 (2.08)	24.33 (2.08)	46.00 (4.00)
Others	20.00 (1.41)	23.00 (1.41)	43.00 (2.83)
**Education** ^b^			
No formal education	22.00 (1.41)	26.00 (0.00) **	48.00 (1.41) **
Primary education	19.73 (2.57)	20.64 (2.66)	40.36 (4.32)
Secondary education	21.21 (2.72)	22.18 (2.76)	43.39 (4.21)
Tertiary education	21.43 (2.49)	22.57 (2.60)	44.00 (4.07)
Occupation sector ^b^			
Unemployed	21.51 (2.26)	22.14 (2.77)	43.65 (3.97)
Government	21.35 (2.55)	22.52 (2.67)	43.87 (4.07)
Private	21.61 (2.64)	22.39 (2.39)	44.00 (3.81)
Self-employed	20.81 (2.76)	22.52 (2.78)	43.33 (4.61)
Pensioner	21.44 (2.75)	22.50 (2.79)	43.94 (4.52)
**Household income bracket** ^c,d^			
B40	21.30 (2.56)	21.03 (2.83) ***	43.33 (4.29) **
M40	21.36 (2.53)	22.70 (2.52)	44.06 (3.82)
T20	21.54 (2.62)	22.91 (2.40)	44.45 (4.10)
**School locality** ^a^			
Urban	21.41 (2.51)	22.58 (2.62)	43.99 (4.05)
Rural	21.29 (2.61)	22.29 (2.70)	43.59 (4.10)
**School type** ^b,e^			
SMK	21.37 (2.42)	22.39 (2.72)	43.76 (4.05)
SBP	21.48 (2.57)	22.70 (2.54)	44.18 (4.00)
SMA	21.17 (2.72)	22.20 (2.71)	43.37 (4.23)
**Personal history of mental illness** ^a^			
Yes	22.29 (2.69)	22.67 (2.89)	44.95 (4.51)
No	21.34 (2.55)	22.46 (2.65)	43.80 (4.06)
**Had known someone with mental illness** ^a^			
Yes	22.23 (2.66) ***	22.89 (2.50) *	45.12 (3.89) ***
No	21.17 (2.49)	22.36 (2.68)	43.53 (4.06)
**Had assisted those with mental** **Illness** ^a^			
Yes	21.83 (2.52) **	22.55 (2.62)	44.38 (3.99) *
No	21.20 (2.54)	22.43 (2.67)	43.64 (4.08)
**Had attended formal psychological first aid training** ^a^			
Yes	21.90 (2.59) *	23.02 (2.34) *	44.92 (3.58) **
No	21.30 (2.54)	22.40 (2.68)	43.70 (4.11)

Note: SD = Standard deviation. ^a^ independent t-test, ^b^ one-way ANOVA, ^c^ Welch’s test ^d^ Malaysian household income stratification. Bottom 40% (B40) (RM4849 and below); middle 40% (M40) (RM4950–RM10959); top 20% (T20) (RM10960 and above). ^e^ School type: national secondary schools (SMK); fully residential schools (SBP); and religious secondary schools (SMA). * mean difference is significant at *p* ≤ 0.05 level, ** mean difference is significant at *p* ≤ 0.01 level, *** mean difference is significant at *p* ≤ 0.001 level.

**Table 4 healthcare-10-01304-t004:** Comparison of mean between sociodemographic and M-LOSS (*n* = 867).

Variables	Causes and Nature Mean (SD)	Risk Factors Mean (SD)	Signs and Symptoms Mean (SD)	Treatment and Prevention Mean (SD)	M-LOSS Mean (SD)
**Role** ^a^					
Parents and caregivers	4.29 (1.84)	4.19 (1.28)	2.68 (0.98)	2.92 (1.06)	14.08 (2.66)
Teachers	4.13 (1.77)	4.33 (1.16)	2.64 (0.96)	2.91 (1.09)	14.00 (2.51)
**Sex** ^a^					
Male	4.24 (1.83)	4.17 (1.25)	2.70 (0.95)	2.98 (1.03)	14.09 (2.58)
Female	4.23 (1.82)	4.27 (1.23)	2.65 (0.98)	2.89 (1.08)	14.04 (2.62)
**Ethnicity** ^b^					
Malay	4.26 (1.79)	4.24 (1.22)	2.67 (1.00)	2.91 (1.07)	14.08 (2.60)
Chinese	3.82 (1.80)	4.39 (1.41)	2.73 (0.91)	2.84 (1.01)	13.78 (2.58)
Indian	4.50 (3.00)	4.75 (1.26)	3.00 (1.16)	2.50 (1.29)	14.75 (3.86)
Other Bumiputera	4.29 (1.08)	4.13 (1.27)	2.54 (0.82)	2.98 (1.11)	13.92 (2.69)
Others	3.64 (2.25)	4.09 (1.70)	2.82 (0.88)	3.36 (0.67)	13.91 (2.70)
**Religion** ^b^					
Islam	4.26 (1.80)	4.23 (1.23)	2.66 (0.99)	2.92 (1.07)	14.08 (2.61)
Christian	3.98 (2.00)	4.18 (1.36)	2.61 (0.81)	2.94 (1.05)	13.71 (2.61)
Buddhist	3.97 (1.73)	4.50 (1.19)	2.76 (0.92)	2.74 (1.05)	13.97 (2.48)
Hindu	5.00 (3.46)	4.33 (1.16)	2.67 (1.16)	2.00 (1.00)	14.00 (4.36)
Others	4.50 (2.12)	4.00 (2.83)	3.00 (1.41)	3.50 (0.71)	15.00 (2.83)
**Education** ^b^					
No formal education	4.00 (0.00)	4.00 (2.83)	2.50 (0.71)	2.50 (0.71)	13.00 (2.83)
Primary education	4.36 (1.69)	4.55 (0.82)	2.55 (0.93)	3.00 (0.89)	14.45 (2.34)
Secondary education	4.24 (1.80)	4.18 (1.31)	2.79 (1.04)	2.93 (1.06)	14.14 (2.67)
Tertiary education	4.23 (1.83)	4.25 (1.22)	2.63 (0.95)	2.91 (1.08)	14.02 (2.60)
**Occupation sector** ^b,c^					
Unemployed	4.38 (1.94)	4.12 (1.25)	2.64 (1.17)	2.97 (1.02)	14.11 (2.86)
Government	4.19 (1.85)	4.27 (1.25)	2.67 (1.95)	2.93 (1.08)	14.07 (2.61)
Private	4.24 (1.68)	4.12 (1.15)	2.65 (0.90)	2.91 (1.05)	13.92 (2.43)
Self-employed	4.48 (1.65)	4.41 (1.17)	2.70 (1.01)	2.83 (1.01)	14.41 (2.45)
Pensioner	3.83 (1.58)	4.11 (1.41)	2.56 (0.92)	2.39 (1.30)	12.89 (2.56)
**Household income bracket** ^b,d^					
B40	4.31 (1.80)	4.24 (1.25)	2.74 (1.01)	2.92 (1.04)	14.21 (2.59)
M40	4.16 (1.83)	4.25 (1.26)	2.63 (0.93)	2.94 (1.07)	13.97 (2.52)
T20	4.27 (1.83)	4.23 (1.13)	2.58 (0.98)	2.82 (1.15)	13.90 (2.93)
**School locality** ^a^					
Urban	4.23 (1.80)	4.26 (1.23)	2.71 (1.00)	2.90 (1.07)	14.11 (2.60)
Rural	4.23 (1.85)	4.21 (1.25)	2.60 (0.94)	2.93 (1.06)	13.98 (2.62)
**School type** ^b,c,e^					
SMK	4.14 (1.83)	4.31 (1.23)	2.67 (0.94) **	3.01 (1.00)	14.13 (2.47) *
SBP	4.35 (1.80)	4.22 (1.21)	2.78 (1.03)	2.88 (1.08)	14.23 (2.71)
SMA	4.20 (1.82)	4.16 (1.28)	2.48 (0.92)	2.81 (1.15)	13.65 (2.64)
**Personal history of mental illness** ^a^					
Yes	4.76 (1.73)	3.95 (1.12)	2.67 (0.91)	2.57 (0.98)	13.95 (2.11)
No	4.22 (1.81)	4.25 (1.24)	2.67 (0.98)	2.92 (1.07)	14.06 (2.62)
**Had known someone with mental illness** ^a^					
Yes	4.44 (1.79)	4.10 (1.22)	2.68 (0.93)	2.97 (1.12)	14.19 (2.66)
No	4.19 (1.82)	4.27 (1.24)	2.66 (0.98)	2.90 (1.06)	14.02 (2.60)
**Had assisted those with mental illness** ^a^					
Yes	4.21 (1.76)	4.19 (1.19)	2.68 (1.00)	2.96 (1.06)	14.04 (2.61)
No	4.24 (1.84)	4.26 (1.25)	2.66 (0.97)	2.90 (1.07)	14.06 (2.61)
**Had attended formal psychological first aid training** ^a^					
Yes	4.36 (1.82)	4.11 (1.22)	2.69 (0.85)	2.92 (1.03)	14.09 (2.51)
No	4.22 (1.82)	4.25 (1.24)	2.66 (0.99)	2.91 (1.07)	14.05 (2.62)

Note: SD = Standard deviation. ^a^ independent t-test, ^b^ one-way ANOVA, ^c^ Welch’s test, ^d^ Malaysian household income stratification. Bottom 40% (B40) (RM4849 and below); middle 40% (M40) (RM4950–RM10959); top 20% (T20) (RM10960 and above). ^e^ School type: national secondary schools (SMK); fully residential schools (SBP); and religious secondary schools (SMA). * mean difference is significant at *p* ≤ 0.05 level, ** mean difference is significant at *p* ≤ 0.01 level.

**Table 5 healthcare-10-01304-t005:** Correlation between age, MAKS-M, and M-LOSS (*n* = 867).

Variables	Age	MAKS-M	M-LOSS
**Age (years)**			
**MAKS-M**	−0.065		
**M-LOSS**	−0.076 *	0.007	

* Correlation is significant *p* ≤ 0.05 level.

## Data Availability

Not applicable.

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
