# Peer review of "The Malay Literacy of Suicide Scale: A Rasch Model Validation and Its Correlation with Mental Health Literacy among Malaysian Parents, Caregivers and Teachers"

_healthcare, 2022, doi:10.3390/healthcare10071304_

Round 1

Reviewer 1 Report

The authors of this paper aim to test the psychometric properties of the Malay-translated version of the LOSS (M-LOSS) in West Malaysia and its correlation to the mental health literacy. This study fills a noticeable gap in validating a culturally appropriate version of a suicide literacy instrument administered to Malaysian parents, caregivers, and teachers of secondary school students. The study could have significant impact on the field of suicide prevention in Malaysia school systems. However, there are some major issues the authors need to address before the consideration of publishing this work. This is such a critical area of study, so this reviewer hopes the authors can update the results which would have a big impact on the field of suicide prevention.

Introduction:

1.      The authors discussed the main aim of the study at the end of the section. However, the rationale on why validating this adapted Malaysian version is important is too generic. What are the key clinical implications of the study other than helping professionals understand the issue and enhancing confidence in providing help? How does improving levels of suicide literacy among parents and teachers helps prevent from adolescent suicidal behaviors? What programs could be potentially developed based off of this study? These are the key questions the authors should address in the introduction section.

2.      In the introduction section, the authors also discussed that suicide does not always indicate an underlying mental illness and tended to ensure that both issues receive equal attention. As suicide literacy is the key measure of interest, from a clinical perspective, in addition to mental health literacy, it is also worth exploring the other risk factors that may contribute to suicide literacy, such as adverse childhood experience.

Materials and Methods

1.      Although the authors did mention the final sample size in the following sections, the final sample size for each analysis should be stated under 2.1 Study Design and Subjects.  Are there any sample power calculation conducted prior to the data collection?

2.      In the Instrument section, it is helpful if the authors add Cronbach’s alpha for each of the LOSS four categories, in addition to the overall one.

3.      I find it difficult to accept treating “Don’t know” as similar to neutral given the fact that Don’t know does not necessarily indicate a middle position. It could also mean lacks enough information to form an opinion or missing data. If there were a very small number of "don't know" responses, you will lose very little information by just coding the "don't know" responses as 3s, which is by far the simplest solution. However, there are up to 33% of responses with Don’t know/Neutral. Unless you have strong theoretical support to treat “don’t know” as 3s for this specific instrument, I suggest the authors recode them as missing and re-run the analysis. To report the internal consistency of the MAKS-M, it is helpful to include it for both Parts A and B separately, in addition to the overall one.

     Results

1.      The results are well presented. However, I found it interesting because in the Rasch Model Analysis, 77% of the respondents completed tertiary education and 68% worked in the government sector. The sample consists of a large portion of parents or teachers with relatively high socio-economic status. Is there any reason? How that affects the generalizability of the study should be discussed.

2.      Can the authors clarify why there are different numbers of respondents for the Rasch Model Analysis and the follow-up analysis with Suicide literacy and mental health literacy? Are there any missing data? How did they address it in all the analyses?

3.      In the Results section, subgroups were compared using the independent t-test and one-way ANOVA. However, what is the rationale for conducting these analyses? Not sure if this answers any research questions stated in the Introduction section, although they look interesting. Also, since the respondents are parents and caregivers, it is even more critical to validate and compare the results from parents with teachers because they are two very distinct groups.

4.      For the post hoc test following one-way ANOVA to compare the mean scores of MAKS-M across groups, the authors used Tukey test, but it is often used when you would like to make pairwise comparisons between group means when the sample sizes for each group are equal, which might not be the case for all the comparisons. The reviewer suggests another approach instead.

5.      I found Table 5 is a bit hard to follow. “aSD” is easy to understand, but is “b P-values” for the correlation coefficient (r)? If so, can you report it next to the r instead?

     Discussion

1.      I found the Discussion section is interesting. However, under 4.2 Mental health literacy and suicide literacy, there are several paragraphs discussing the factors associated with both literacy scores. It echoes what I am suggesting above that, the authors should clearly state one of the aims is also to investigate the correlates or factors that contribute to the literacy scores.

2.      Again, I feel obligated to suggest the authors discuss the different results for parents/caregivers versus teachers because they are very different.

Author Response

Hello Reviewer 1,

We would like to thank you for your thorough feedback and suggestions. Attached is the word file [Review Report (Reviewer 1)] containing the authors' responses to your comments.

Thank you.

Reviewer 2 Report

I would like to thank the authors for a thorough and well considered piece of research.  The introduction builds a good rationale for the study.  Methods section is clear and explained well with a good level of detail.  Analysis and results are clear and appropriate. Discussion is well considered.  I just have some very minor comments below;

page 1, line 37 -  'which could intensify for those with existing mental health problems' - the sentence doesn't read very well with the term 'intensify'.  perhaps 'which could exacerbate existing mental health problems' would be better? 

page 1, line 39 - I don't think I agree that there are 'few factors associated with a higher risk of suicidal behaviours' - please re-word to remove 'few' from this point. 

page 2, line 46 - it's not clear how less screening for depression at care visits would lower the infection risk. 

page 2 line 87 - should be 'a' cross-sectional design rather than 'the' cross-sectional design. 

page 2 line 88 - should be 'comprised' rather than 'comprises'

page 15 line 392 - remove 'is' from 'the current study is aimed..'

Author Response

Hello Reviewer 2,

We would like to thank you for your thorough feedback and suggestions. Attached is the word file [Review Report (Reviewer 2)] containing the authors' responses to your comments.

Thank you.

Round 2

Reviewer 1 Report

The authors addressed my comments appropriately.

There is one minor suggestion: In the procedure section, I suggest the authors generate a flow chart to show the steps and sample size for each step. That will help readers better understand the design as well as why the sample size differs across the analyses.  

Author Response

Dear reviewer,

We thank you for the constructive feedback. We have added study flowchart to illustrate the procedure of this study better, as suggested.

Thank you. 
